# *Pseudomonas aeruginosa* induces p38MAP kinase-dependent IL-6 and CXCL8 release from bronchial epithelial cells via a Syk kinase pathway

**Matthew S. Coates**[1] *, **Eric W. F. W. Alton**[1], **Garth W. Rapeport**[1,2], **Jane C. Davies**[1,3], **Kazuhiro Ito**[1,2]

**1** National Heart and Lung Institute, Imperial College London, London, United Kingdom, **2** Pulmocide Ltd, London, United Kingdom, **3** Department of Paediatric Respiratory Medicine, Royal Brompton Hospital, London, United Kingdom

* m.coates12@imperial.ac.uk

## Abstract

*Pseudomonas aeruginosa* (Pa) infection is a major cause of airway inflammation in immuno-compromised and cystic fibrosis (CF) patients. Mitogen-activated protein (MAP) and tyrosine kinases are integral to inflammatory responses and are therefore potential targets for novel anti-inflammatory therapies. We have determined the involvement of specific kinases in Pa-induced inflammation. The effects of kinase inhibitors against p38MAPK, MEK 1/2, JNK 1/2, Syk or c-Src, a combination of a p38MAPK with Syk inhibitor, or a novel narrow spectrum kinase inhibitor (NSKI), were evaluated against the release of the proinflammatory cytokine/chemokine, IL-6 and CXCL8 from BEAS-2B and CFBE41o- epithelial cells by Pa. Effects of a Syk inhibitor against phosphorylation of the MAPKs were also evaluated. IL-6 and CXCL8 release by Pa were significantly inhibited by p38MAPK and Syk inhibitors ($p < 0.05$). Phosphorylation of HSP27, but not ERK or JNK, was significantly inhibited by Syk kinase inhibition. A combination of p38MAPK and Syk inhibitors showed synergy against IL-6 and CXCL8 induction and an NSKI completely inhibited IL-6 and CXCL8 at low concentrations. Pa-induced inflammation is dependent on p38MAPK primarily, and Syk partially, which is upstream of p38MAPK. The NSKI suggests that inhibiting specific combinations of kinases is a potent potential therapy for Pa-induced inflammation.

**Data Availability Statement:** All relevant data are within the manuscript and its Supporting information files.

## Introduction

*Pseudomonas aeruginosa* (Pa) is an opportunistic pathogenic bacterium, which is normally found in soil or aqueous environments [1]. Pa infection is of particular importance in the CF lung, where early acquisition is associated with a reduction in lung microbiota diversity and an accelerated reduction in lung function [2, 3]. Pa-induced lung inflammation is driven by host cell production of cytokines and chemokines, which are induced by multiple Pa virulence components, either on the cell surface, such as lipopolysaccharide, flagella and pili [4–7], or

**Funding:** Matthew S. Coates received funding for this project from Respivert Ltd. The funders had no role in study design, data collection and analysis, decision to publish, or preparation of the manuscript. During the period of the research Garth W. Rapeport and Kazuhiro Ito were co-founders/employed by Pulmocide Ltd. The funder provided support in the form of salaries for authors [GWR and KI], but did not have any additional role in the study design, data collection and analysis, decision to publish, or preparation of the manuscript. The specific roles of these authors are articulated in the 'author contributions' section.

**Competing interests:** I have read the journal's policy and the authors of this manuscript have the following competing interests: Previously Matthew S. Coates was employed by Respivert Ltd. Garth W. Rapeport and Kazuhiro Ito were co-founders and employees of Respivert Ltd. During the period of the research G. W. Rapeport and K. Ito were co-founders and employees of Pulmocide Ltd. This does not alter our adherence to PLOS ONE policies on sharing data and materials.

secreted, including type III secretion system products [8], quorum sensing molecules [9, 10] and pyocyanin [11]. Of particular importance are the Pa-induced chemokine CXCL8, which is instrumental in neutrophil migration to the site of infection [12], and the cytokine IL-6, which is involved in the release of acute phase proteins and immune cell differentiation [13].

One of the earliest interactions between inhaled pathogens such as Pa and the host is with bronchial epithelial cells, therefore making these an important line of defence and immune/inflammatory cell. Pa pathogen-associated molecular patterns (PAMPs) are detected by epithelial cell transmembrane cellular pattern recognition receptors (PRRs). PAMPS such as TLR4, TLR2 and TLR5, which detect lipopolysaccharide (LPS) [14], pili and flagellin [6, 15], respectively, induce intracellular signal pathways resulting in the release of proinflammatory cytokines [16]. In the CF lung the potent chemokine CXCL8 [12] attracts large numbers of neutrophils to the site of infection; the latter can comprise up to 95% of the luminal cellular population, compared to approximately 5% in healthy individuals [17]. The high numbers of neutrophils in the airways induce a proinflammatory cycle with inhibition of normal innate and adaptive host defence responses and resultant biofilm formation [18].

Through phosphorylation of their targets, protein kinases finely control activation of specific intracellular signal cascades such as inflammatory pathways [19]. MAP kinases, which in humans are grouped into extracellular signal-regulated kinases (ERKs), c-Jun N-terminal kinases (JNKs) and p38 mitogen-activated kinases (p38MAPK), are integral to intracellular signal cascades regulating cell proliferation, differentiation and death [20]. P38MAPK is known to be strongly activated by stress signals and plays a role in immune responses as well as cell survival and differentiation [21]. Epithelial cells expressing the cystic fibrosis transmembrane conductance regulator (*CFTR*) gene with the Phe508del mutation—the most common mutation in patients with CF—show hyper-activity of p38MAPK in response to Pa [22–24], which may be a result of endoplasmic reticulum stress and therefore sensitisation of the cells to microbial stimuli [25]. In Phe508del CF cells, MAP kinases, like ERK and p38MAPK, are thought to be hyper-reactive to Pa materials due to an altered sensitivity to reactive oxygen species, and are implicated in the high baseline activation of NF-κB and AP-1 in CF epithelial cells [22, 26]. JNK kinase, which is activated by cellular stress and is involved in AP-1 activation has previously been shown to be involved in the inflammatory response of bronchial epithelial cells to the bacterial component, LPS [27, 28].

Tyrosine kinases, which specifically phosphorylate tyrosine residues within target proteins, can be subdivided into receptor tyrosine kinases and non-receptor tyrosine kinases, the latter including Sarcoma kinase (Src) family kinases and spleen tyrosine kinase (Syk) [29]. One of the earliest events that takes place after detection of PAMPs by TLRs is phosphorylation of tyrosine on downstream molecules. Thus, non-receptor tyrosine kinases must be closely linked to TLR activation [29]. Syk kinase is an important component of PRR signalling and through it a multitude of signal cascades are activated, including the MAP kinases and NF-κB [30]. Recently, the use of the Syk inhibitor R406 has been shown to inhibit p38MAPK and ERK phosphorylation, as well as reducing inflammatory cytokine release from human monocyte cells [31]. Src kinases have previously been shown to be instrumental in LPS-induced cytokine, IL-6 and TNFα release from macrophages, and this involved a level of control over p38MAPK and ERK 1/2 MAP kinases [32].

MAP kinase inhibitors have been trialled for the treatment of inflammatory diseases such as rheumatoid arthritis [33] and chronic obstructive pulmonary disease (COPD) [34–36], and also for Alzheimer's [37, 38], neuropathic pain [39] and cancers such as non-small cell lung cancer [40, 41]. More specifically, p38MAPK inhibitors have been assessed in rheumatoid arthritis and Crohn's [42] but their anti-inflammatory effects have only been transient, possibly due to cells using alternative pathways as a redundancy strategy once p38MAPK has been

blocked [43]. Alternative targets for anti-inflammatory therapies are required, including the inhibition of the upstream tyrosine kinases such as Src and Syk. Currently Src kinase inhibitors are used for the treatment of Philadelphia positive chronic myeloid leukaemia [44, 45], and Alzheimer's diseases [46]. Syk inhibitors have been trialled for rheumatoid arthritis [47, 48] and are used for the treatment of immune thrombocytopenia [49, 50]. Targeting Syk kinase has been suggested for the treatment of Pa-induced inflammation [30], but there are no Syk inhibitors available currently for respiratory inflammation. The use of an NSKI targeting p38MAPK, Src and Syk, and a combination of single kinase inhibitors, has been described by Knobloch *et al.* [51], showing efficacy even in steroid-resistant models.

However, to date no multiple kinase analysis of Pa-induced inflammation from bronchial epithelial cells has been completed. Full understanding of the pathways involved in inflammation could help future development of anti-inflammatory therapies of clinical benefit for CF patients. We have therefore compared, head-to-head, specific kinase inhibitors' abilities to inhibit Pa-induced IL-6 and CXCL8 from epithelial cells, and investigated the signal pathway through kinase phosphorylation measurement and compound combinations. Finally, we used a novel NSKI to investigate how blocking multiple specific kinases could be a potent potential anti-inflammatory therapy.

## Materials and methods

### Cell and bacteria culture

The simian virus 40 (SV-40) immortalised bronchial epithelial BEAS-2B cell line (ATCC® CRL-9609™, ATCC, Manassas, VA, USA) was continuously maintained at 37˚C, 5% $CO_2$ in medium made of equal volumes of LHC-8 medium without gentamicin (12679–015, Life Technologies, Paisley, UK) and Roswell Park Memorial Institute medium (RPMI)– 1640 with 15 mM L-glutamine and phenol red (11875–093, Life Technologies) (growth medium). CFBE41o- cells, expressing either WT- or Phe508del-CFTR, donated by E. J. Sorscher, University of Alabama, are stably transfected with either Phe508del- or WT-CFTR cDNA using TranzVector™, and express the appropriate form of the CFTR protein [52]. The cells were continuously maintained at 37˚C, 5% $CO_2$ in minimum essential media (MEM) with 15nM L-glutamine, phenol red (31095–029, Life Technologies), supplemented with 10% [$^v/_v$] heat inactivated foetal bovine serum (FBS). Cells were seeded in either 96, or 6 well tissue culture treated plates (non-pyrogenic, polystyrene, 3590 and 3516, Corning, NY, USA) at 3 x $10^5$ cells/ml, 100 µl per well, or 5 x $10^5$ cell/ml, 2 ml per well, respectively, in growth medium and incubated overnight at 37˚C, 5% $CO_2$.

*Pseudomonas aeruginosa* (Pa) strain PAO1 (ATCC®15692™) was grown on Pseudomonas specific agar, made up of *Pseudomonas* agar base (CM0559, Oxoid, Basingstoke, UK), with 1% glycerol and Pseudomonas C-N selective supplement (SR0102, Oxoid), at 37˚C. Prior to infection of cells, a single colony was streaked on a fresh agar plate and incubated at 37˚C for 24 hours, after which a colony was collected and suspended in Miller's Luria broth (12795–027, Invitrogen, Carlsbad, CA, USA) and grown at 37˚C, with shaking, 200 RPM, overnight. The Pa were pelleted and re-suspended in cell growth media, the bacterial concentration (CFU/ml) was then calculated by optical density (OD) at 600 nm, and checked by serially dilution and plating, which showed that an OD of 1 was equivalent of approximately 7 x $10^8$ CFU/ml.

### Test article preparation

Commercially available kinase inhibitors were synthesised and supplied by Sygnature Discovery Ltd (Nottingham, UK). Compounds were diluted in DMSO to achieve concentrations 200-fold higher than the desired final concentration, to achieve a constant concentration of

**Table 1. Kinase inhibitors used, with their primary target and molecular weight.**

| Kinase Inhibitor | Primary Target | Molecular Weight |
|---|---|---|
| SB203580 | p38MAPK α, β [53] | 377 |
| BIRB796 | p38MAPK α, δ, γ [54] | 528 |
| SP600125 | JNK 1/2 [55] | 220 |
| PD98059 | MEK 1/2 [56] | 267 |
| BAY 61–3606 | Syk [57] | 463 |
| Dasatinib | c-Src [58] | 488 |
| RV1088 | p38MAPK α, γ, Src, Syk [59] | 593 |

DMSO of 0.5% [$^v/_v$] in all experiments. The compounds used and their primary targets are described below in Table 1.

## Cytokine stimulation and measurement

Cell growth media was replaced on the cells with fresh LHC-8, or MEM–with no FBS–media as appropriate, 200 μl per well. The test articles were prepared and added to appropriate wells, 1 μl per well, and vehicle (DMSO) added to the Pa infection alone and no infection control wells and incubated for two hours at 37˚C, 5% $CO_2$.

Recombinant human tumour necrosis factor alpha (TNFα) was pre-diluted to 50 ng/ml in LHC-8 media and 50 μl was added to appropriate wells to give a final concentration on 10 ng/ml as previously described [60]. LPS from *Pseudomonas aeruginosa* 10 (L9143, Sigma Aldrich, St Louis, MO, USA) was pre-diluted to 50, 500 or 5,000 ng/ml in LHC-8 medium and 50 μl was added to appropriate wells to give a final concentration of 10, 100 or 1,000 ng/ml as previously described [61]. Plain LHC-8 media, 50 μl, was added to non-treatment wells. The BEAS-2B cells were incubated with the appropriate stimulants for four hours at 37˚C, 5% $CO_2$, after which 200 μl of cell free supernatant was collected and stored at -20˚C.

Pa was diluted to 1.25 x $10^8$ CFU/ml in appropriate media and added to appropriate wells, in triplicate, 50 μl per well, to give a final concentration of 2.5 x $10^7$ CFU/ml. The Pa and cells were incubated together, 37˚C, 5% $CO_2$, for one hour, after which gentamicin was added to all wells at a concentration of 100 μg/ml [5] to prevent over-growth of Pa and subsequent epithelial cell death. The cells were incubated for a further four hours, 37˚C, 5% $CO_2$, then 200 μl of cell-free media was collected and stored at -20˚C for cytokine analysis.

The concentrations of the proinflammatory cytokine and chemokine IL-6 and CXCL8 in the collected cell-free supernatant were measured using sandwich ELISA duosets (DY208 and D206, R&D Systems, Abingdon, UK). The kits were used as per the manufacturer's instructions, with the cell supernatant being diluted in reagent diluent prior to addition to the ELISA plate. Concentrations were calculated from a standard curve, percent inhibitions and $IC_{50}$ values determined using GraphPad Prism (GraphPad Software, La Jolla, CA, USA). For statistical analysis any samples below the lower end of the standard curve were assigned the value that would be next on the standard curve.

## Cell viability

3-(4,5-Dimethyl-2-thiazolyl)-2,5-diphenyl-2H-tetrazolium bromide (MTT) was used as a measure of cell viability, with the reduction of MTT to formazan crystals as a relative measure of viable cells [62]. BEAS-2B cells seeded in 96 well plates were treated with test articles for two hours, followed by addition of Pa and gentamicin as previously described and incubated at

$37°C$, 5% $CO_2$. Cell supernatant was removed and replaced with 100 μl of MTT, 0.5 mg/ml, (0793-1g, VWR, Lutterworth, UK) in LHC-8 media and incubated for 1 hour at $37°C$, 5% $CO_2$. MTT solution was removed and replaced with 50 μl of neat DMSO and incubated for 15 minutes at RT with shaking (70 RPM). Optical density (OD) of wells was measured at 550 nm, OD of wells not containing cells was subtracted from all other wells, and percent viability of the Pa/kinase inhibitor treated cells compared to Pa/vehicle treated cells was calculated.

## Western blot of phosphorylated kinases

BEAS-2B cells grown in 6 well plates were stimulated with Pa at a final concentration of 2.5 x $10^7$ CFU/ml in 1 ml of LHC-8 medium. Cells and bacteria were incubated, $37°C$, 5% $CO_2$, together for two hours. This time point was chosen as our previous optimisation showed peak HSP27 phosphorylation at two hours PI [63], gentamicin was not added as the short incubation period meant that the Pa did overgrow and kill the epithelial cells. The medium was then removed, and the cells washed with ice cold Dulbecco's PBS with protease and phosphatase inhibitor cocktail (PPI; MSSAFE SIGMA, Sigma Aldrich). Cells were then scraped on ice in the presence of radioimmunoprecipitation assay (RIPA) buffer containing PPI before duplicate wells were combined and incubated on ice for 30 minutes with vigorous mixing every 10 minutes. The samples were centrifuged at 13,000 x *g* for 15 minutes at 4°C and the cell lysate collected. Total protein levels were measured using a Bradford assay (500–2005, Bio-Rad, Watford, UK) and 20 μg of each sample was taken forward. Samples were reduced using NuPAGE™ LDS sample buffer (NP0007, Life Technologies) and NuPAGE™ sample reducing agent (NP0009, Life Technologies) and run on a NuPAGE™ Novex 4–12% Bis-Tris gel (NP0321, Life Technologies) for 45 minutes at 120 mA and 200 V to separate the proteins. The proteins were transferred to a nitro-cellulose membrane and detected using appropriate primary antibodies (anti-Hsp27, 1:1000, ab2790, Abcam, Cambridge, UK, anti-phosphorylated Hsp27 (phospho S78), 1:2000, ab32501, Abcam, anti-ERK 1/2 (c-9), 1:200, sc-514302, Santa Cruz biotechnology, Dallas, TX, USA, anti-phosphorylated ERK (E-4), 1:200, sc-7383, Santa Cruz biotechnology, anti-JNK (D-2), 1:500, sc-7345, Santa Cruz biotechnology, anti-phosphorylated JNK (G-7), 1:500, sc-6254, Santa Cruz biotechnology) and species specific HRP-conjugated secondary antibodies (polyclonal goat anti-rabbit, P0448, Agilent, Santa Clara, CA, USA polyclonal goat anti-mouse, P0447, Agilent) followed by enhanced chemiluminescence prime western blotting reagent (RPN2236, GE Healthcare, Amersham, UK). The protein bands were imaged using a Syngene G:Box Chemi XRQ (Syngene, Cambridge, UK) camera, and the band intensities were quantified using Syngene Gene Tools software. The levels of phosphorylated protein were corrected to the levels of total protein in each sample.

## Statistical analysis

All statistical analyses were performed using GraphPad Prism V8.4. Comparisons of two sets of data were carried out using the Wilcoxon signed-rank test if paired, and Mann-Whitney U test if non-paired. Comparisons of multiple paired data sets were carried out using a Friedman's test with Dunn's multiple comparisons. The null hypothesis was rejected at $p < 0.05$ and indicated on figures with asterisks. The $IC_{50}$ values were calculated using Graphpad Prism, plotting a three-parameter curve, from which the 50% inhibition was extrapolated. Synergy between compounds was calculated using the Chou-Talalay method using CompuSyn software (ComboSyn Inc., Paramus, NJ), which was used to calculate combination indexes at the concentrations required to inhibit 90% of cytokine release.

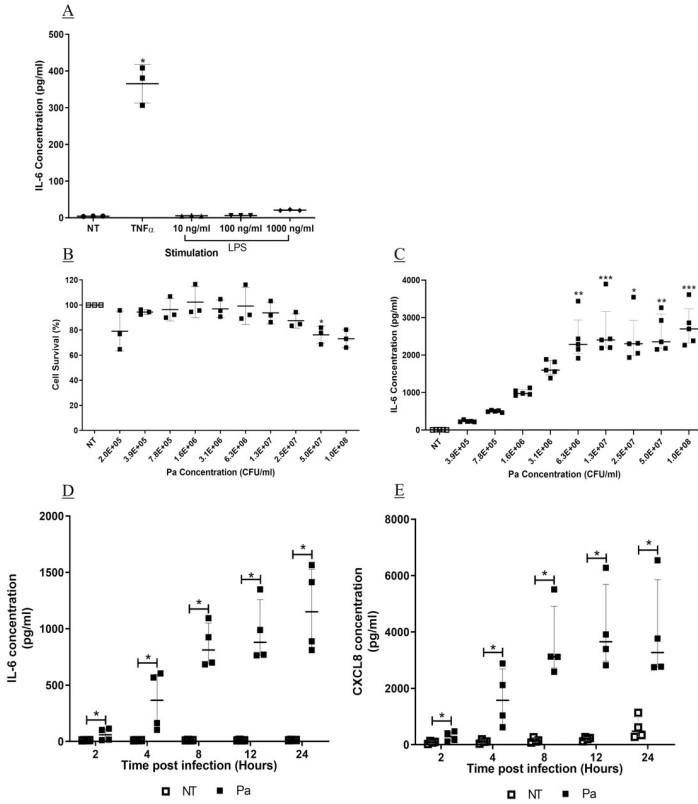

**Fig 1. Stimulation of IL-6 and CXCL8 from BEAS-2B cells.** BEAS-2B cells were stimulated with (A) TNFα at 10 ng/ml or LPS at 10, 100 or 100 ng/ml for four hours. Cell free supernatant was collected, and IL-6 measured by sandwich ELISA (n = 3, mean ± SD). (B) BEAS-2B cells were infected with Pa at varying concentrations for one hour, followed by addition of gentamicin at 100 μg/ml and a further 24 hours incubation. Cell viability was measured by MTT assay (n = 3, mean ± SD). (C) Cell free supernatant from cells infected with Pa for 24 hours was analysed for IL-6 by sandwich ELISA (n = 5, median with IQR). BEAS-2B cells were infected with Pa at 2.5 x $10^7$ CFU/ml for one hour, followed by addition of gentamicin at 100 μg/ml. Supernatant was collected at 2, 4, 8, 12 or 24 hours later. (D) IL-6 and (E) CXCL8 were measured by sandwich ELISA (n = 4, median with IQR). (A-C) Stimulants were compared to NT using Friedman test with Dunn's correction. (D-E) At each time point, Pa treatments were compared with NT using the Mann-Whitney U test (* = p<0.05** = p<0.01, *** = p<0.001).

## Results

### Stimulation of proinflammatory cytokines from bronchial epithelial cells

To confirm that the BEAS-2B cells could produce proinflammatory cytokines, cells were stimulated with the Pa cell wall component LPS and the human cytokine TNFα. LPS stimulation did not result in a significant release of IL-6 at the four hour time point at any concentration tested (20.9 ± 1.4 pg/ml at 1000ng/ml) compared with non-stimulated (NT) cells (4.4 ± 0.6 pg/ml, Fig 1A); LPS was, therefore, not considered an appropriate stimulant in this model (Fig 1A). In contrast, TNFα showed a significant induction of IL-6 (365.4 ± 52.6 pg/ml, p<0.05, Fig 1A), confirming that these cells responded to TNFα and that BEAS-2B cells were an appropriate cell line for further inflammatory investigation.

We further assessed whole, live Pa as a potential stimulant of BEAS-2B cells. Cells were stimulated for one hour with varying concentrations of Pa, followed by addition of gentamicin, to prevent overgrowth of Pa, and a further 24 hours incubation. Cell survival was significantly reduced at a concentration of 5 x $10^7$ CFU/ml, (76.2 ± 6.8% cell survival, p<0.05, Fig 1B).

However, at $2.5 \times 10^7$ CFU/ml there was no significant reduction in cell survival compared with the NT control ($87.5 \pm 5.9\%$, Fig 1B). There was little release of IL-6 into the supernatant of NT wells, (median, 9.4 pg/ml (IQR, 3.1–9.4)), compared with a concentration related increase following Pa stimulation. Significant induction of IL-6 compared with NT was detected at Pa concentrations of $6.3 \times 10^6$ CFU/ml, (2285.4 pg/ml (IQR, 2155.4–2433.4)), and above ($p < 0.01$, Fig 1C). A Pa concentration of $2.5 \times 10^7$ CFU/ml was used for further investigations.

The optimal infection period was also investigated, with assessment of Pa-induced IL-6 and CXCL8 at 2, 4, 8, 12 and 24 hours post infection. The levels of IL-6 from NT cells remained similar at all time points. Pa infection induced IL-6 in a time-dependent manner increasing from 2 to 24 hours post infection with a peak 97.4-fold induction (IQR, 62.0–125.0) (Fig 1D). Significant induction compared with NT was seen at all time points ($p < 0.05$, Fig 1D). Conversely, CXCL8 from NT cells increased over time (Fig 1E), and concentrations from Pa stimulated cells peaked at 12 hours post infection; the peak induction compared with NT was seen at eight hours post infection (31.0 fold (IQR, 12.6–47.7) (Fig 1E). As seen with IL-6 release, there was a significant induction of CXCL8 at all time points investigated ($p < 0.05$, Fig 1E); therefore, to allow for pre-incubation with kinase inhibitors a stimulation period of four hours was chosen moving forward.

## Pa-induced IL-6 and CXCL8 are highly dependent on p38MAPK and Syk kinases

BEAS-2B cells were treated with single kinase inhibitors at concentrations that have previously been shown to give complete inhibition of enzyme activity in a cell-free assay. Of the MAP kinase inhibitors tested, the p38MAPK inhibitor, SB203580, showed significant inhibition of Pa-induced IL-6 (92.3% (IQR, 91.3–93.0, $p < 0.05$, Fig 2A) suggesting that p38MAPK is integral to the signalling of Pa-induced inflammation. The other MAP kinase inhibitors SP600125, JNK 1/2, and PD98059, MEK 1/2, did not show significant inhibitions of IL-6 (Fig 2A). We, therefore, conclude that p38MAPK is the dominant MAP kinase in Pa-induced IL-6.

In addition to the p38MAPK inhibitors, the Syk inhibitor BAY 61–3606 also showed significant inhibition of IL-6 induction (94.3% (IQR, 93.5–96.0), $p < 0.01$, Fig 2A). The c-Src inhibitor, dasatinib, however, produced no inhibition (Fig 2A). Therefore, of the tyrosine kinases investigated only Syk kinase appears to be involved in Pa-induced IL-6.

As seen with IL-6, p38MAPK was found to be the dominant MAP kinase in CXCL8 induction signalling, but also there was significant inhibition by a JNK 1/2 inhibitor ($p < 0.05$, Fig 2B). Syk kinase was also found to have a significant role in CXCL8 release ($p < 0.01$, Fig 2B).

Cell viability after compound treatment was assessed using an MTT assay, showing that none of the compound treatments resulted in a significant reduction in cell viability (Fig 2C). Therefore, it can be assumed that the inhibitions of IL-6 and CXCL8 were due to kinase inhibition, not a result of reduced cell number.

As the p38MAPK and Syk inhibitors showed significant inhibition of both Pa-induced IL-6 and CXCL8 release these compounds were taken forward for full concentration investigation. Of the p38MAPK inhibitors BIRB796, an inhibitor of the α, δ and γ isoforms, showed significantly greater potency against IL-6 ($IC_{50}$ $4.97 \times 10^{-4}$ µg/ml (IQR, $3.68–7.54 \times 10^{-4}$) than SB203580, an α and β inhibitor ($IC_{50}$ $5.07 \times 10^{-3}$ µg/ml (IQR, $3.91–5.79 \times 10^{-3}$) ($p < 0.01$, Fig 3A and 3B). This shows that BIRB796 is 11-fold more potent than SB203580, but previously it has shown less than 2-fold greater potency against p38MAPK α, in enzyme assays [53, 54]. However, at a concentration of BIRB796 that resulted in only 82.1% (IQR, 79.1–85.0) inhibition of IL-6 (Fig 3C) the compound achieved complete inhibition of p38MAPK α and β

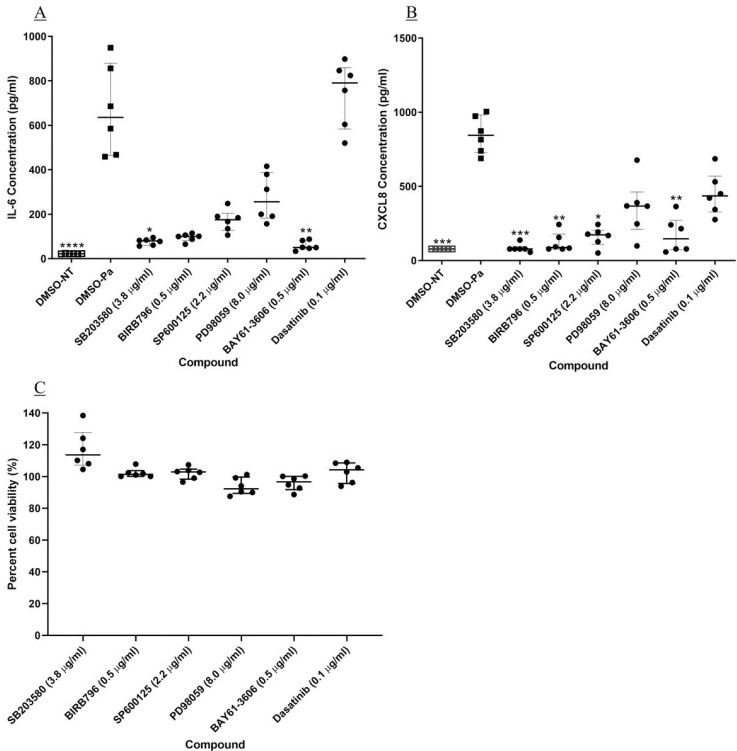

**Fig 2. Head to head comparison of kinase inhibitors against Pa-induced IL-6 and CXCL8 release, and cell viability.** BEAS-2B cells underwent a two-hour pre-treatment with kinase inhibitors at a single concentration known to completely inhibit the target kinase enzyme activity. The cells were stimulated with Pa at $2.5 \times 10^7$ CFU/ml for one hour followed by addition of gentamicin at 100 µg/ml and a further four-hour incubation. The concentration of (A) IL-6 and (B) CXCL8 in the cell-free supernatant was measured by sandwich ELISA. (C) BEAS-2B cell viability was measured after compound treatment and Pa infection, using an MTT assay; cell viability was compared to cells treated with DMSO and Pa. Friedman test with Dunn's multiple comparisons was used to compare each kinase inhibitor with vehicle treatment. n = 6, showing median with IQR (* = p<0.05, ** = p<0.01 and *** = p<0.001). Compound concentrations: SB203580–3.8 µg/ml; BIRB796–0.5 µg/ml; SP600125–2.2 µg/ml; PD98059–8.0 µg/ml; BAY 61–3606–0.5 µg/ml and Dasatinib– 0.1 µg/ml.

activity (p<0.01, Fig 4A and 4B). BAY 61–3606, a Syk inhibitor showed significant inhibition of IL-6 only at 1 µg/ml, (p<0.05, Fig 3C), with an IC$_{50}$ value of 0.36 µg/ml (IQR, 0.26–0.41). Similar results were seen when investigating the involvement of the kinases in Pa-induced CXCL8 release (Fig 3D–3F). Activity of SB203580 suggests that p38MAPK α and β have an important role in both Pa-induced IL-6 and CXCL8 release and the difference between SB203580 and BIRB796 suggests that the δ and γ isoforms may also be involved.

## A Syk inhibitor blocks Pa-induced HSP27, but not ERK or JNK phosphorylation

In a human monocyte cell line it has been shown that a Syk inhibitor can prevent Pa-induced phosphorylation of p38MAPK, ERK2 and JNK kinases [31], suggesting that Syk kinase is upstream of the MAP kinases, and could control their activity. Therefore, the effects of the Syk inhibitor BAY 61–3606 on the phosphorylation of HSP27, a surrogate of p38MAPK α and β activity, ERK and JNK kinases in Pa-stimulated BEAS-2B cells was investigated.

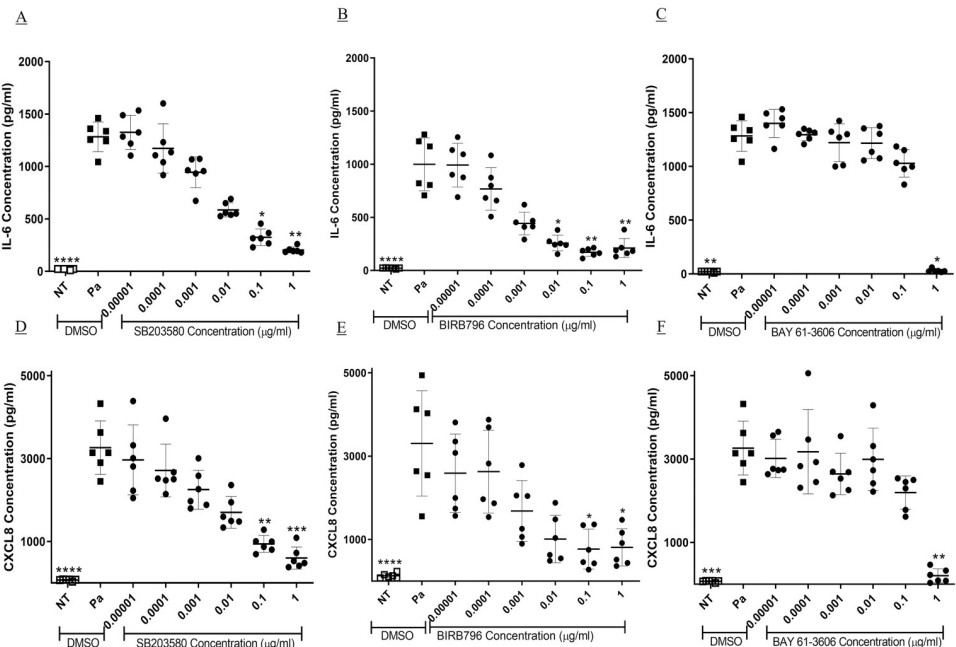

**Fig 3. Concentration dependent inhibition of Pa-induced IL-6 and CXCL8 release by selected kinase inhibitors.**
BEAS-2B cells underwent a two-hour pre-treatment with 10-fold dilutions of (A and D) SB203580, (B and E) BIRB796 and (C and F) BAY 61–3606. The cells were stimulated with Pa at 2.5 x $10^7$ CFU/ml for one hour followed by addition of gentamicin at 100 μg/ml and a further four-hour incubation. The concentration of (A, B and C) IL-6 and (D, E and F) CXCL8 in the cell free supernatant was measured by sandwich ELISA. Friedman test with Dunn's multiple comparisons was used to compare each kinase inhibitor with vehicle treatment. n = 6, showing median with IQR (* = $p < 0.05$, ** = $p < 0.01$ and *** = $p < 0.001$).

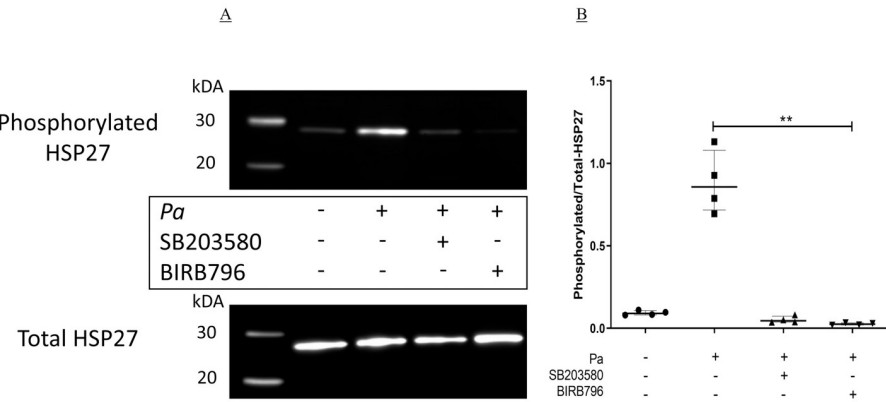

**Fig 4. Inhibition of Pa-induced phosphorylation of HSP27 by p38MAPK inhibitors.** BEAS-2B cells in six well plates were pre-incubated with either DMSO (NT), SB203580 or BIRB796 at 1 μg/ml for two hours at 37°C, 5% $CO_2$. The cells were then stimulated with Pa at 2.5 x $10^7$ CFU/ml for two hours at 37°C, 5% $CO_2$, after which whole cell protein was collected and levels of phosphorylated- and total-HSP27 were measured by Western blot. (A) Representative image of the four experiments. (B) Band intensity was determined using Syngene Gene Tools software and the levels of phosphorylated-HSP27 were corrected to the levels of total-HSP27. A Friedman test with Dunn's multiple comparison was used to compare Pa treatment alone with all other conditions. n = 4 showing median with IQR (** = $p \leq 0.01$).

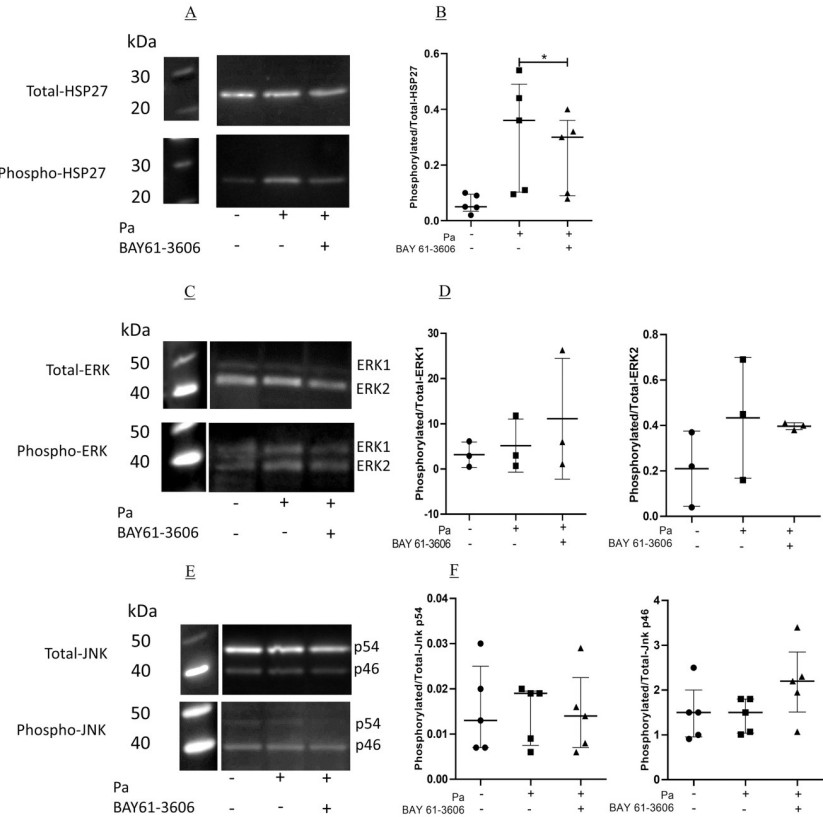

**Fig 5. Inhibition of MAP kinase activity by BAY 61–3606.** BEAS-2B cells in six well plates were pre-incubated with either DMSO, or BAY 61–3606 at 1 μg/ml for two hours at 37˚C, 5% $CO_2$. The cells were then incubated with Pa at 2.5 x $10^7$ CFU/ml for two hours at 37˚C, after which whole cell protein was collected. Western blots were carried out to measure the levels of total and phosphorylated (A) HSP27, (C) ERK and (E) JNK. (B, D and F) Band intensities were calculated using Syngene Gene Tools and phosphorylated protein levels were corrected to total protein. BAY 61–3606 treatment was compared to DMSO using a one-tailed Wilcoxon sign ranked test. (B) and (F), n = 5, median with IQR, (D) n = 3, mean ± SD. Images are representative of the repeats conducted (* = p<0.05).

BAY 61–3603 significantly inhibited the phosphorylation of HSP27, 32.2% (18.9–34.0, median and IQR, p<0.05, Fig 5A and 5B) indicating that the compound inhibited p38MAPK α and β activity. At the time point tested there was no significant phosphorylation of ERK1 or ERK2, and no inhibition by BAY 61–3606 (Fig 5C and 5D). When measuring JNK phosphorylation it was seen that there was no significant activation of the kinase by Pa, and there was no inhibition by BAY 61–3606 (Fig 5E and 5F). Therefore, in bronchial epithelial cells, Syk kinase is upstream of, and can determine activity of p38MAPK during Pa stimulation.

## Synergistic effects of inhibitors of p38MAPK and Syk kinases against Pa-induced IL-6 and CXCL8

Inhibition of Pa-induced IL-6 by combinations of BIRB796 and BAY 61–3606 were investigated; synergy of the compounds would suggest that the p38MAPK and Syk kinase are involved in separate signalling pathways. Epithelial cells were pre-treated with combinations of BIRB796 and BAY 61–3606 at a set concentration ratio followed by stimulation with Pa. Synergy of compounds was assessed by the Chou-Talalay method, with a combination index (CI) of less than one indicating synergy [64].

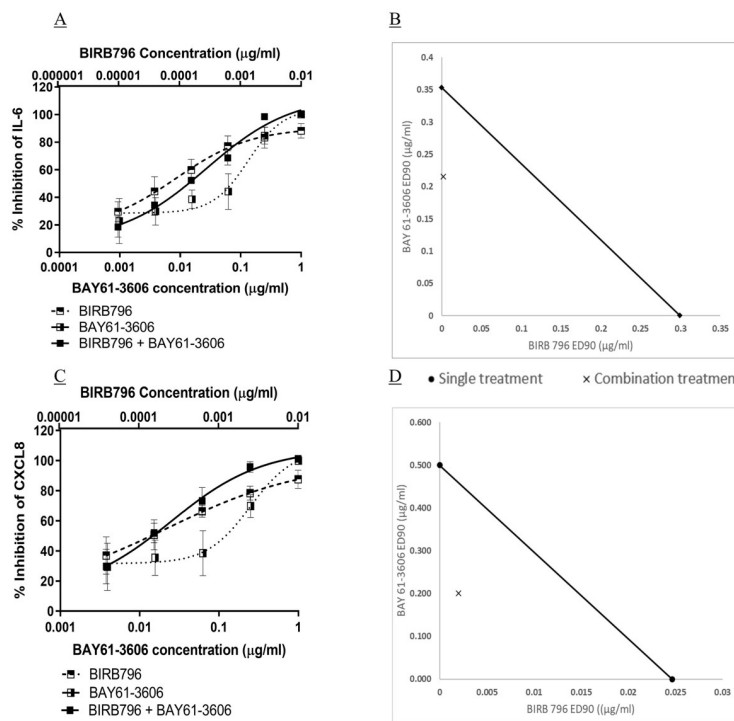

**Fig 6. Synergy of p38MAPK and Syk inhibition against Pa-induced IL-6 and CXCL8 release.** BEAS-2B cells underwent a two-hour pre-treatment with four-fold dilutions of either BIRB796 or BAY 61–3606, or a combination of the two. The cells were stimulated with Pa at $2.5 \times 10^7$ CFU/ml for one hour followed by addition of gentamicin at 100 µg/ml and a further four-hour incubation. The concentration of (A) IL-6 and (C) CXCL8 in the cell free supernatant were measured by sandwich ELISA, n = 3, showing mean ± SD. Synergy was calculated using the Chou-Talalay method using CompuSyn software [64]. Isobolograms of the $ED_{90}$ concentrations for the compound concentrations against (B) IL-6 and (D) CXCL8 were drawn using Microsoft Excel. The $ED_{90}$ of each compound applied alone is plotted on the x and y axes and joined with a line. The $ED_{90}$ values of each compound when used in combination are plotted as an X. An X on the line would indicate an additive effect; X on the left of the line would indicate synergy (as in these experiments), and to the right would indicate compound antagonism.

Combinations of BIRB796 and BAY 61–3606 showed an increase in maximal inhibition of IL-6 by BIRB796 and maintained the higher efficacy at lower combined concentrations than either compound alone (Fig 6A) but did not show more potency than BIRB796 alone. Synergy was seen when determining the $ED_{90}$ values of compound combinations compared with compounds alone, with a CI of 0.5 ± 0.01 (mean ± SD, Fig 6B). The $IC_{50}$ of BIRB796 when in combination with BAY 61–3606 was $1.70 \times 10^{-4}$ µg/ml ± $2.30 \times 10^{-5}$ and the $IC_{50}$ of BAY 61–3606, in combination, was 0.017 µg/ml ± 0.002 (Fig 6A). Inhibition of CXCL8 using a combination of BIRB796 and BAY 61–3606 showed similar results to IL-6, with a higher inhibition, maintained at lower combined concentrations, than either compound alone ($ED_{90}$ CI value of 0.6 ± 0.3, Fig 6C and 6D). These results indicate that that both Pa-induced IL-6 and CXCL8 release are dependent on both p38MAPK and Syk kinase signalling and that these kinases are involved in signal pathways independently of each other. RV1088, a novel, NSKI that specifically targets p38MAPK, Syk and c-Src was used to further investigate the effects of inhibiting multiple kinases simultaneously against Pa-induced IL-6 and CXCL8 release. RV1088 showed potent inhibition of both IL-6 and CXCL8, with significant inhibition at concentrations as low

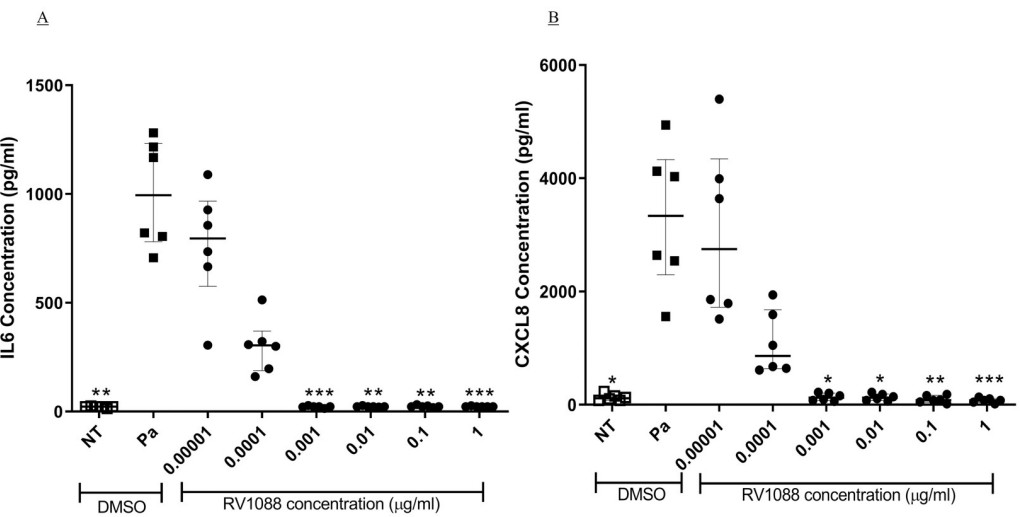

**Fig 7. Inhibition of IL-6 and CXCL8 by a novel narrow spectrum kinase inhibitor.** BEAS-2B cells underwent a two-hour pre-treatment with 10-fold dilutions of RV1088. The cells were stimulated with Pa at $2.5 \times 10^7$ CFU/ml for one hour followed by addition of gentamicin at 100 µg/ml and a further four-hour incubation. The concentration of (A) IL-6 and (B) CXCL8 in the cell free supernatant was measured by sandwich ELISA. Friedman test with Dunn's multiple comparisons was used to compare each kinase inhibitor with vehicle treatment. n = 6, showing median with IQR (* = $p < 0.05$, ** = $p < 0.01$ and *** = $p < 0.001$).

as 0.001 µg/ml, and $IC_{50}$ values of $2.84 \times 10^{-5}$ µg/ml (IQR, $2.62$–$5.05 \times 10^{-5}$) and $4.14 \times 10^{-5}$ µg/ml (IQR, $3.04$–$5.65 \times 10^{-5}$), respectively ($p < 0.05$, Fig 7A and 7B). This makes RV1088 six-fold and 600-fold more potent than BIRB796 and BAY 61–3606, respectively, when used in combination against Pa-induced IL-6 from BEAS-2B cells.

## Inhibition of p38MAPK and Syk kinases prevent Pa-induced IL-6 release from a CF cell line

The paired epithelial cell line, CFBE41o-, expressing either WT or Phe508del CFTR, were stimulated with Pa, as previously described, to investigate whether inhibitors of p38MAPK, Syk or an NSKI could reduce the inflammatory response from CF cells.

Pa showed almost two-fold induction of IL-6 from Phe508del CFTR expressing epithelial cells (NT; 15.9 pg/ml (IQR, 10.7–18.7), Pa; 28.0 pg/ml (IQR, 17.7–38.5) Fig 8A–8C). The p38MAPK inhibitor, BIRB796, significantly inhibited Pa induced IL-6 release at 0.01 µg/ml ($p < 0.05$), with an $IC_{50}$ of $5.16 \times 10^{-5}$ µg/ml (IQR, $4.89$–$6.27 \times 10^{-5}$, Fig 8A). The Syk inhibitor, BAY 61–3606, showed an $IC_{50}$ value of $0.23 \pm 0.22$ µg/ml (mean $\pm$ SD, Fig 8B). The NSKI, RV1088, showed significant inhibition of Pa-induced IL-6 at 0.01 µg/ml ($p < 0.01$), with an $IC_{50}$ value of $4.02 \times 10^{-5}$ µg/ml (IQR, $2.63$–$5.17 \times 10^{-5}$, Fig 8C). Similar results were seen when inhibiting Pa-induced IL-6 from the WT CFTR expressing cells (Fig 8D–8F), and were similar in the BEAS-2B cells. Therefore, p38MAPK and Syk kinase also have a dominant role in Pa-induced inflammatory signalling in Phe508del CFTR expressing epithelial cells.

## Discussion

Using a head-to-head comparison of kinase inhibitors in bronchial epithelial cells this study has shown that p38MAPK is the dominant MAP kinase in Pa-induced IL-6 and CXCL8 release, with both JNK 1/2 and ERK 1/2 also contributing activity. Further, we show that Pa-

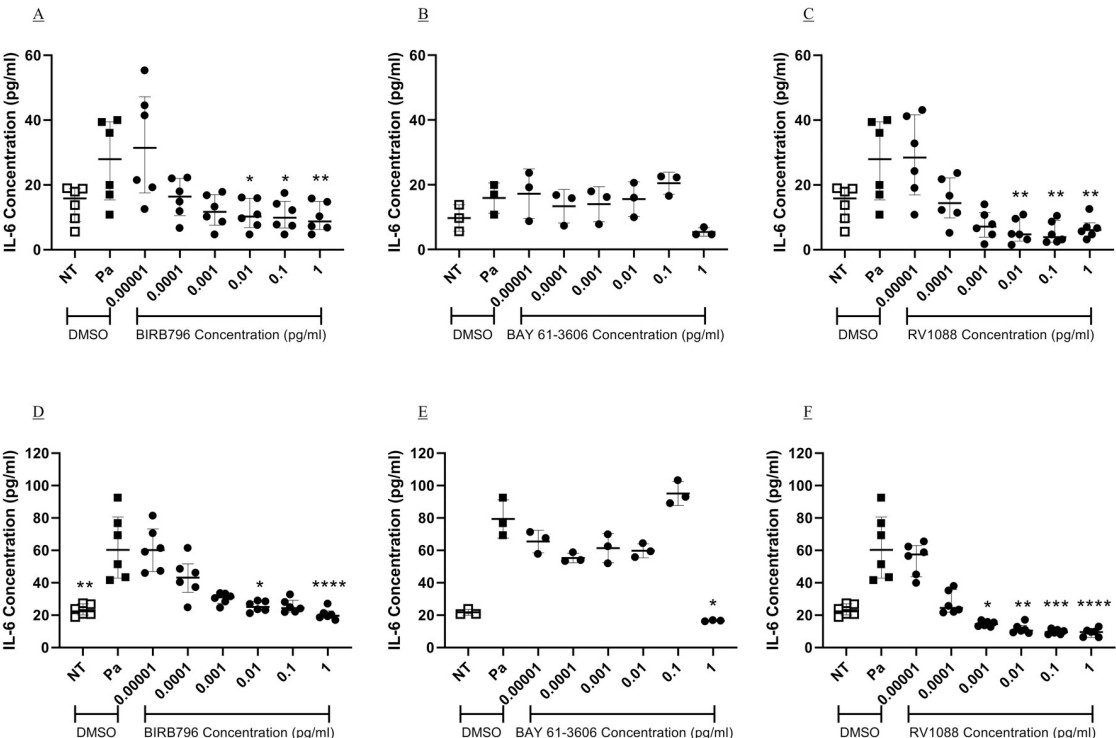

**Fig 8. Inhibition of Pa-induced IL-6 from bronchial epithelial cells expressing Phe508del- and WT-CFTR.** CFBE41o- cells expressing either Phe508del (A-C) or WTCFTR (D-F) underwent a two-hour pre-treatment with 10-fold dilutions of BIRB796 (A and D), BAY 61–3606 (B and E) and RV1088 (C and E). The cells were stimulated with Pa at 2.5 x $10^7$ CFU/ml for one hour followed by addition of gentamicin at 100 μg/ml and a further four-hour incubation. The concentration of IL-6 in the cell free supernatant was measured by sandwich ELISA. Friedman test with Dunn's multiple comparisons was used to compare each kinase inhibitor with vehicle treatment. n = 6 for BIRB796 and RV1088, showing median with IQR, n = 3 for BAY 61–3606, showing mean and SD (* = p<0.05, ** = p<0.01 and *** = p<0.001, **** = p<0.0001).

induced IL-6 and CXCL8 release are both highly dependent on Syk tyrosine kinase activity, whereas c-Src kinase showed only a partial role in CXCL8 release. Activity of Syk kinase was found to be upstream of p38MAPK and had a contributory role in its activation. The use of a novel NSKI indicates that targeted inhibition of multiple selected kinases is a potential potent therapy for Pa-induced inflammation. The dominant roles of p38MAPK and Syk kinases were also confirmed in bronchial epithelial cells expressing Phe508del CFTR.

Previous studies have shown that BEAS-2B are an appropriate cell line for investigating epithelial cell cytokine response, and were significantly stimulated by TNFα. However, they are not well stimulated by certain TLR agonists such as LPS, due to a lack of the co-receptor cluster of differentiation 14 [61]. Having confirmed this, rather than using individual TLR agonists, whole live bacteria were used for further investigation in this study. Optimisation of Pa infection of BEAS-2B cells identified that a bacterial concentration of 2.5 x $10^7$ PFU/ml significantly induced IL-6 release with little cell death and an incubation period of four hours was optimal for both IL-6 and CXCL8 measurement.

The current study did not show greater Pa-induced IL-6 from epithelial cells expressing Phe508del CFTR, compared with WT. This agrees with previous research showing little difference between IL-6 induction from WT and CF primary epithelial cells and cell lines by multiple stimulants [65, 66]. This however, is counter to the higher Pa-induced IL-6 secretion by

Phe508del CFTR expressing cells reported by Berube *et al.* [22]. However, that study used different cell lines, with the CF and WT cells originally from different donors and which were, therefore, not paired. Also, Berube *et al.* used sterile culture filtrates rather than whole cells, the study, therefore, not being directly comparable to the live infection used in the current study.

The activity of p38MAPK in Pa-induced signalling is of particular interest as in bronchial epithelial cells expressing Phe508del CFTR hyper-activation of this kinase in response to Pa culture filtrate has been shown to result in higher IL-6 release compared with cells expressing WT CFTR [22]. Also, in Phe508del CFTR-expressing cells the incomplete folding of the CFTR protein results in altered intracellular protein homeostasis and endoplasmic reticulum stress. This sensitises the cellular immune response, which leads to a greater response to bacterial stimuli, via p38MAPK signalling [25]. Previously an antimicrobial compound, TP359, was found to inhibit both Pa-induced CXCL8 and IL-6 from airway epithelial cells, due to inhibition of p38MAPK phosphorylation [67]. Ruffin *et al.* showed that differentiated bronchial epithelial cells expressing Phe508del CFTR released less CXCL8 in response to Pa diffusible material after treatment with the CFTR corrector/potentiator combination Vx-809/Vx-770 [68]. This reduction of CXCL8 release was associated with reduced p38MAPK phosphorylation, indicating a link between CFTR function and p38MAPK activity. Berube *et al.* have shown that the higher levels of p38MAPK phosphorylation seen in CF patient airway epithelial cells not only led to high transcription of cytokine mRNA, but also increased the mRNA stability [22]. Therefore, p38MAPK may be the dominant MAP kinase in Pa-induced inflammation by prolonging the time mRNA is available for translation and thereby increasing cytokine release.

However, p38MAPK inhibitors have been investigated for the treatment of Crohn's disease and rheumatoid arthritis but, to date, none have been proven successful [69]. Clinical trials have shown that reduction of markers such as C-reactive protein were only transient, with levels returning to baseline even with continued treatment. Also, the concentrations of inhibitors required to produce their effects were shown to result in toxicity of the liver [42, 70–72]. *In vitro* investigation has shown that possible redundancy pathways are used by cells once p38MAPK is blocked, which are controlled by kinases upstream of the MAPKs [73, 74]. P38MAPK may play a regulatory role in the activation of other kinases. In macrophages, p38MAPK inhibition was shown to upregulate JNK and ERK activity under inflammatory conditions [75] and in fibroblasts p38MAPK inhibition can upregulate TAK1, thereby increasing activity of JNK, and IkB which could increase NF-kB activity [76]. Therefore, other signalling pathways are of importance in Pa-induced IL-6 and CXCL8.

As well as p38MAPK, the other MAPKs, JNK 1/2 and ERK 1/2, showed trends towards inhibition of Pa-induced IL-6 and CXCL8 release, but did not reach statistical significance for both cytokines. JNK 1/2 has previously been shown to be involved in TNFα, but not flagellin-induced CXCL8 from epithelial cells [6] ERK 1/2 have been implicated in the intrinsic inflammation of CF [26] and are known to be involved in Pa and flagellin-induced CXCL8 and IL-6 release via NF-κB activation [6, 22, 23]. However, the use of live Pa in this study means that PAMPS other than flagellin could be inducing the signal cascades and could explain the activity we observed.

Of the two tyrosine kinases investigated, Syk kinase was shown to be involved in Pa-induced IL-6 release whereas Src was not. Syk kinase is well known to be expressed in haematopoietic cells, but has also been shown to be expressed in airway epithelial cells [77]. Syk kinase is integral to inflammation in the monocytic THP-1 and lung epithelial H292 cell lines; in the latter the Syk inhibitor R406 inhibited Pa-induced TNFα and IL-1β release [31]. These data show that Syk kinase is towards the start of the cell inflammatory signalling pathway and that it can control the phosphorylation of p38MAPK, ERK 2, JNK and IκBα [78]. Syk kinase is

known to associate with cell surface receptors such as TLRs and ICAM-1 in response to stimuli such as Pa, TNFα and rhinovirus [30, 79], which can activate four separate signal cascades, including the p38MAPK pathway [77]. Here we have shown that a Syk inhibitor, BAY 61–3606, significantly inhibited both IL-6 and CXCL8 by over 90%. However, the compound showed low potency, which could be a result of Syk's upstream position in the signalling pathway, such that low level activity may be amplified through the signal cascade. We show that BAY 61–3606 significantly inhibited p38MAPK α/β activity as evidenced by the reduction in HSP27 phosphorylation, which concurs with the work by Alhazmi *et al.* and suggesting that Syk kinase is upstream of p38MAP kinase, and several other reports showing Syk inhibition led to inhibition of phosphorylation of p38MAPK [80–82]. However, given inhibition was not complete there are likely to be other pathways controlling p38MAPK activation. BAY 61–3606 showed no inhibition of the phosphorylation of JNK 1/2 or ERK 1/2 kinases in this system, which could be due to the fact that there was little phosphorylation detected; in the future further time points could be investigated to determine if phosphorylation could be detected. Although, the lack of JNK 1/2 phosphorylation is not congruent with other results in this study, which showed a reduction in Pa-induced IL-6 and CXCL8 by SP600125. At the concentration tested (10 μM), required for complete inhibition of JNK 1/2 activity, off target effects have been noted including inhibition of JNK 3 and even p38MAPK activity [83].

Combinations of the p38MAPK inhibitor, BIRB796, and the Syk inhibitor, BAY 61–3606, showed synergistic inhibition of both IL-6 and CXCL8 when comparing the concentrations required to inhibit 90% of the cytokine/chemokine release. This synergy concurs with the data showing that inhibition of Syk kinase was only able to inhibit p38MAPK α/β activity by approximately 30% and that a portion of both kinase pathways must be independent of each other. In addition, Syk kinase is known to be involved in four separate pathways, including via p85 and AKT to activate NF-κB and potentially inducing cytokine release [84], bypassing MAP kinase signalling. However, Syk kinase is vital for host defence signalling and its inhibition in neutrophils can reduce their ability to kill bacteria via phagocytosis and extracellular trap formation [85]. Further, trials of a Syk inhibitor have shown increased occurrence of upper respiratory tract infections compared with placebo controls [86]. Therefore, future studies will need to consider the possible side effects of Syk inhibition in non-target cells.

An NSKI targeting p38MAPK, Src kinases and Syk kinase, RV1088, has been shown to be potent at inhibiting inflammation in human airway smooth muscle cells from COPD patients [51]. RV1088 was an effective inhibitor of TNFα- and LPS-induced CXCL8 release, whereas BIRB796 was not, suggesting that multiple kinases are separately involved in these signalling cascades. However, in the current model, although RV1088 showed increased potency compared to BIRB796, both compounds showed significant inhibition of both IL-6 and CXCL8 at higher concentrations. Previously, both RV1088 and BIRB796 showed similar potencies against TNFα- and LPS-induced GM-CSF from COPD human airway smooth muscle cells [51], suggesting that in p38MAPK dominant signalling this may be the most important kinase of the three targeted. The synergy results reported here indicate that inhibiting multiple kinases results in lower compound concentrations being needed to achieve the high inhibition of cytokine release that are likely to be required for an anti-inflammatory drug. For inhaled drugs, sub-nanomolar potency is optimal to prevent off target toxicity, the latter not helped by only ~20% of any compound reaching the lungs following nebulised delivery [87]; synergy between kinase inhibitors is, therefore, of particular value. The inhibition of multiple kinases, including those sited more proximally in the signalling pathways, could be an important component in combatting the transient effects seen by specific p38MAPK inhibitors and providing a more long term solution for chronic inflammatory diseases exemplified by persistent Pa infections.

We acknowledge several potential limitations. BEAS-2B and CFBE41o- cells are immortalised bronchial epithelial cells and may, therefore, differ in their phenotype as compared with primary respiratory epithelial cells. Primary bronchial epithelial cells grown in air liquid interface cultures have a more similar transcriptional profile to the native airway epithelium in unstimulated conditions [88]. However, these cells could not be purchased in the quantities required for the current investigation. The use of the paired CFBE41o- cells expressing WT or Phe508del CFTR has allowed us to show that the signalling pathways are similar in CF and WT cells, although these only represent a single mutation, so future studies should use primary cells with different CFTR-mutations. Small molecule kinase inhibitors are not completely specific inhibitors of their target kinases and siRNA, or other knockdown studies will provide useful additional verification of the inflammatory signalling pathways. The use of a single time point in the measurement of kinase-phosphorylation is a limitation, as activation of kinases is transient, and can happen quickly after stimulation. Therefore, future studies would carry out full time course experiments to determine the time of peak phosphorylation of each kinase of interest. It would have also been of interest to determine the levels of Pa-induced Syk kinase phosphorylation. Investigating p38MAPK inhibitors impact on Syk phosphorylation would also have given further evidence that Syk was upstream in the signalling pathway, and so this will be completed in future studies. We also applied gentamicin to prevent the overgrowth of the Pa and subsequent death of the epithelial cells during the infection period rather than complete elimination of live bacteria. Consequently, all bacteria were not confirmed to be dead and remaining extracellular or invasive intracellular live bacteria might affect cytokine release additionally. We would like to address this in a future study. The addition of gentamicin also may prevent the bacteria from releasing the extracellular virulence factors that are released during an infection. Therefore, future studies would address this by stimulation with Pa toxins and proteases, or sterile bacterial culture filtrate, potentially allowing longer incubation periods without cell death. Clearly within the *in vivo* environment there are further interactions, not just between bacteria and the epithelium, but also with immune cells such as macrophages and neutrophils which play a role in the inflammatory response. As always, *in vivo* animal model studies will be helpful to confirm the roles of specific kinases in Pa-induced cytokine release. Finally, we used the laboratory adapted Pa strain, PAO1. There is large variation in the phenotype of clinical strains of Pa, including biofilm formation, LPS structure and the release of virulence factors and also Pili, flagella mutants [89–91]. Previously, a difference in the signalling cascade induced by mucoid and non-mucoid strains of Pa has been seen, with mucoid strains inducing corticosteroid-resistant inflammation, driven by TLR2 signalling [92]. It will, therefore, be helpful to expand the range of acute and chronic Pa isolates.

This study has shown that Pa-induced IL-6 and CXCL8 from bronchial epithelial cells, both WT and CF, is highly dependent on p38MAPK and the tyrosine kinase Syk; Syk kinase is likely upstream of p38MAPK. Synergy of the Syk inhibitor with a p38MAPK inhibitor indicates that Syk must activate pathways other than p38MAPK and/or p38MAPK can be activated by multiple upstream kinases. This synergy, as well as the potent inhibition of both IL-6 and CXCL8 by an NSKI, suggest that targeting a few select kinases may be a potential new mechanism for anti-inflammatory therapy in the context of Pa infections in CF and other patients.

## Supporting information

**S1 File.**
(PDF)

## Acknowledgments

Permission to use the compound RV1088 was given by RespiVert Ltd, a wholly owned subsidiary of Janssen Biotech, Inc. The compound BIRB796 and RV1088 were provided by Sygnature Discovery Ltd. on behalf of RespiVert Ltd. CFBE41o- cells were a donated for the research by E. J. Sorscher at the University of Alabama.

## Author Contributions

**Conceptualization:** Matthew S. Coates, Jane C. Davies, Kazuhiro Ito.

**Formal analysis:** Matthew S. Coates.

**Investigation:** Matthew S. Coates.

**Methodology:** Matthew S. Coates.

**Supervision:** Eric W. F. W. Alton, Garth W. Rapeport, Jane C. Davies, Kazuhiro Ito.

**Writing – original draft:** Matthew S. Coates.

**Writing – review & editing:** Matthew S. Coates, Eric W. F. W. Alton, Garth W. Rapeport, Jane C. Davies, Kazuhiro Ito.

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
