## [Decision Letter · Decision Letter 0]

22 Oct 2020

PONE-D-20-23493

Pseudomonas aeruginosa induces p38MAP kinase-dependent IL-6 and CXCL8 release from bronchial epithelial cells via a Syk kinase pathway

PLOS ONE

Dear Dr. Coates,

Thank you for submitting your manuscript to PLOS ONE. After careful consideration, we feel that it has merit but does not fully meet PLOS ONE’s publication criteria as it currently stands. Therefore, we invite you to submit a revised version of the manuscript that addresses the points raised during the review process.

Your manuscript, titled "Pseudomonas aeruginosa induces p38MAP kinase-dependent IL-6 and CXCL8 release from bronchial epithelial cells via a Syk kinase pathway", has now been reviewed by two experts in the field. As you can see, both reviewers felt the manuscript was interesting but they had a number of concerns related to the quality of the figures and the lack of certain key data needed to support your overall conclusions. If you feel that you can address the concerns detailed below, I will entertain a revision of the manuscript.

We look forward to receiving your revised manuscript.

Kind regards,

Shawn B Bratton, Ph.D.

Academic Editor

PLOS ONE

Journal Requirements:

2.PLOS ONE now requires that authors provide the original uncropped and unadjusted images underlying all blot or gel results reported in a submission’s figures or Supporting Information files. This policy and the journal’s other requirements for blot/gel reporting and figure preparation are described in detail at https://journals.plos.org/plosone/s/figures#loc-blot-and-gel-reporting-requirements and https://journals.plos.org/plosone/s/figures#loc-preparing-figures-from-image-files. When you submit your revised manuscript, please ensure that your figures adhere fully to these guidelines and provide the original underlying images for all blot or gel data reported in your submission. See the following link for instructions on providing the original image data: https://journals.plos.org/plosone/s/figures#loc-original-images-for-blots-and-gels.

3.Thank you for stating the following in the Financial Disclosure section:

[Matthew Coates received funding for this project from Respivert Ltd. The funders had no role in study design, data collection and analysis, decision to publish, or preparation of the manuscript.].   

We note that one or more of the authors are employed by a commercial company: Pulmocide Ltd

Reviewers' comments:

Reviewer's Responses to Questions

**Comments to the Author**

1. Is the manuscript technically sound, and do the data support the conclusions?

Reviewer #1: Partly

Reviewer #2: Yes

2. Has the statistical analysis been performed appropriately and rigorously? 

Reviewer #1: Yes

Reviewer #2: Yes

3. Have the authors made all data underlying the findings in their manuscript fully available?

Reviewer #1: Yes

Reviewer #2: Yes

4. Is the manuscript presented in an intelligible fashion and written in standard English?

Reviewer #1: No

Reviewer #2: Yes

5. Review Comments to the Author

Reviewer #1: In this manuscript by Coates et al., the authors conclude that Pseudomonas aeruginosa (Pa) infection triggers IL-6/CXCL8 release in bronchial epithelial cells through Syk and p38 MAPK signaling pathways. Overall, the data support this conclusion, although there are several remaining issues with the manuscript.

General Concerns:

1. The figures are very low resolution and almost impossible to read. It makes it very difficult to closely examine or interpret the data within this manuscript. These figures need to be vector graphic files to rectify the pixelation and blurriness.

2. It’s hard to tell given the poor resolution, but indicators of statistical significance appear on some but not all of the graphs. Including this on all graphs would increase clarity.

3. I do not see the value of Table 2 or 3. What exactly do they provide that the graphs (obviously once the resolution is improved) do not? An additional graph showing % inhibition could be included instead of Table 3 if truly desired. The tables also lack legends clarifying which experiments they are tied to.

4. Some small typos/errors remain in the manuscript, the text should be carefully edited prior to publication

Specific Concerns:

1. The intro refers to the common CF mutation Phe508del but does not reference the specific gene (CFTR).

2. I cannot read the axis labels for Fig. 2, but regardless the concentrations of the inhibitors should be stated in the figure legends. It appears to say 1nM for BAY 61-3606? If so, that is equal to ~0.5ng/ul, which is dramatically lower than the reported IC50 in Fig. 3C (~350ng/ul). This apparent discrepancy is not addressed.

3. The authors state “However, at a concentration of BIRB796 that did not give complete inhibition of IL-6 or CXCL8 there was complete inhibition of p38MAPK α and β activity (p<0.01, Figs 4A and B).” This statement is unclear because 1ug/ml was used for both compounds in Fig. 4, yet this concentration was highest dose used in Fig. 3 and showed almost complete inhibition of IL-6 and CXCL8 release.

4. The authors also state “Activity of SB203580 suggests that p38MAPK α and β have an important role in both Pa-induced IL-6 and CXCL8 release and the difference between SB203580 and BIRB796 suggests that the δ and γ isoforms are also involved.” This statement is not supported by the data in Fig. 4. All the data in Fig. 3 show is that BIRB796 is a more potent inhibitor of IL-6 and CXCL8 release. This could simply be because it’s a better inhibitor of p38α.

5. In order to determine the effect of BAY 61-3606 on p38MAPK activity, p38 activation should be analyzed with a phospho-p38 antibody. It would also be informative to determine whether Pa results in Syk activation using a phospho-Syk antibody. Does p38 inhibition reciprocally affect Syk activation?

6. The protein ladder lanes in Fig 5 are not labeled.

7. The IC50s of RV1088 should be determined and compared with co-treatment with BIRB796/BAY 61-3606.

8. RV1088 and/or BIRB796/BAY 61-3606 should also be tested in CFTR (Phe508del) expressing bronchial epithelial cells to evaluate its therapeutic potential.

Reviewer #2: Minor issues:

Please introduce “MAP” abbreviation in abstract.

Format, Table 1, last row.

Preferred observe/present data: graphs rather than tables.

LPS, Bacterial source and provider/company info?

Please discuss LPS-Src / MAPK Activation/Potential reasons for negative LPS-induced IL-6 expression in relation to TLRs.

Major Issues:

Pa-

a) Strain’s phenotype/ use of Pili, flagella mutants or clinical strains is recommended.

b) Cell exposure consistency: One hour or 2 (line 189)?

c) The role of potential Pa’s Toxins and Proteases should also be addressed for longer incubation times.

Gentamycin:

a) The purpose for use needs to be mentioned.

b) Affects loose or adhered bacteria, which takes several hours, but not cytosolic/intercellular Pa. These issues need to be addressed.

Western blot:

a) “kinase phosphorylation assays” may be interpreted as an enzymatic assay than detecting phosphorylated kinase by Western blot.

b) Signaling events are switched on quickly and often last a few minutes. A such, western blot analysis of target intermediate molecules and inhibitors effect should have a time/dose curve.

MTT assay

• Need to be performed for Pa, inhibitors, and in combination.

• Viability protocol is not comparable to assay set up.

• Need to observe and report cell morphology (fluorescence microscopy) to confirm MTT data

6. PLOS authors have the option to publish the peer review history of their article (what does this mean?). If published, this will include your full peer review and any attached files.

Reviewer #1: No

Reviewer #2: No

---

## [Author Response · Author response to Decision Letter 0]

4 Dec 2020

Journal Requirements

[Comment 1]

We have ensured that the manuscript and figures meet the PLOS ONE’s style requirement, including file names.

[Comment 2]

PLOS ONE now requires that authors provide the original uncropped and unadjusted images underlying all blot or gel results reported in a submission’s figures or Supporting Information files.

Included in our resubmission is a file called “S1_raw-images,” which includes the unadjusted images that underlie all of the blot results. 

[Comment 3]

Author contributions and funding statements have been reviewed and the new funding statement is below. 

Funding statement:

Matthew Coates received funding for this project from Respivert Ltd. The funders had no role in study design, data collection and analysis, decision to publish, or preparation of the manuscript. During the period of the research Garth W. Rapeport and Kazuhiro Ito were co-founders/employed by Pulmocide Ltd. The funder provided support in the form of salaries for authors [GWR and KI], but did not have any additional role in the study design, data collection and analysis, decision to publish, or preparation of the manuscript. The specific roles of these authors are articulated in the ‘author contributions’ section.

Competing interests statement: 

I have read the journal's policy and the authors of this manuscript have the following competing interests: Previously M. Coates was employed by Respivert Ltd. G. Rapeport and K. Ito were co-founders and employees of Respivert Ltd. During the period of the research G. Rapeport and K. Ito were co-founders and employees of Pulmocide Ltd. This does not alter our adherence to PLOS ONE policies on sharing data and materials.

Reviewer #1

General concerns

[Comment 1]

The figures are very low resolution and almost impossible to read. It makes it very difficult to closely examine or interpret the data within this manuscript. These figures need to be vector graphic files to rectify the pixelation and blurriness.

We are a little puzzled by this. All figures were saved as TIF files with a resolution of 600 DPI as suggested in the manuscript requirements, so I am not sure if there was an error in the uploading process. As some of the figures have been updated for resubmission all of them have been re-saved as high resolution TIF files, and I will double check quality after uploading. 

[Comment 2]

It’s hard to tell given the poor resolution, but indicators of statistical significance appear on some but not all of the graphs. Including this on all graphs would increase clarity.

Where statistical significance was found it has been indicated on appropriate figures with an asterisk. Where no asterisk is seen there was no statistical significance; this is now stated in the “Statistical analysis” section.

[Comment 3] 

I do not see the value of Table 2 or 3. What exactly do they provide that the graphs (obviously once the resolution is improved) do not? An additional graph showing % inhibition could be included instead of Table 3 if truly desired. The tables also lack legends clarifying which experiments they are tied to.

As per the recommendations Tables 2 and 3 have been removed from the manuscript and the appropriate data added to the text.

[Comment 4]

Some small typos/errors remain in the manuscript, the text should be carefully edited prior to publication.

Thank you; the manuscript has been re-proofed by all authors to ensure that errors have been corrected. 

Specific Concerns

[Comment 1]

The intro refers to the common CF mutation Phe508del but does not reference the specific gene (CFTR).

We thank the reviewer for picking up this oversight; the CFTR gene has now been introduced into the text prior to the Phe508del mutation in lines 72-74, “Epithelial cells expressing the cystic fibrosis transmembrane conductance regulator (CFTR) gene with the Phe508del mutation - the most common mutation in patients with CF -…”. 

[Comment 2]

I cannot read the axis labels for Fig. 2, but regardless the concentrations of the inhibitors should be stated in the figure legends. It appears to say 1nM for BAY 61-3606? If so, that is equal to ~0.5ng/ul, which is dramatically lower than the reported IC50 in Fig. 3C (~350ng/ul). This apparent discrepancy is not addressed.

Apologies again for lack of clarity of compound concentration on figure 2. For continuity throughout the paper all concentrations have been changed to µg/ml, so that figures can be more easily compared. Also, compound concentrations have been added to the legend of Figure 2. BAY 61-3606 was tested at 0.5 µg/ml as a single concentration, so higher than the IC50 calculated of 0.36 µg/ml. 

[Comment 3]

The authors state “However, at a concentration of BIRB796 that did not give complete inhibition of IL-6 or CXCL8 there was complete inhibition of p38MAPK α and β activity (p<0.01, Figs 4A and B).” This statement is unclear because 1ug/ml was used for both compounds in Fig. 4, yet this concentration was highest dose used in Fig. 3 and showed almost complete inhibition of IL-6 and CXCL8 release.

We interpreted that because BIRB796 at 1ug/ml gave approximately 80% inhibition of IL-6 and CXCL8 the difference between this and the complete inhibition of p38MAPK α and β activity is important, as it indicates other pathways must be active. This has been clarified in line 341 with the statement “However, at a concentration of BIRB796 that resulted in only 82.1 % (IQR, 79.1-85.0) inhibition of IL-6 (Fig 3C) the compound achieved complete inhibition of p38MAPK α and β activity (p<0.01, Figs 4A and B).”

[Comment 4]

The authors also state “Activity of SB203580 suggests that p38MAPK α and β have an important role in both Pa-induced IL-6 and CXCL8 release and the difference between SB203580 and BIRB796 suggests that the δ and γ isoforms are also involved.” This statement is not supported by the data in Fig. 4. All the data in Fig. 3 show is that BIRB796 is a more potent inhibitor of IL-6 and CXCL8 release. This could simply be because it’s a better inhibitor of p38α.

We agree that the statement in the manuscript was not clear why we concluded the different isoforms of p38MAPK to be involved. The statement in lines 338-341, “This shows that BIRB796 is 11-fold more potent than SB203580, but previously is has shown less than 2-fold greater potency against p38MAPK α in enzyme assays (53,54).” clarifies that the little difference in p38MAPK α enzyme inhibitory activity between SB203580 and BIRB796 is unlikely to explain the difference in potencies seen in the cytokine release from cells. 

[Comment 5]

In order to determine the effect of BAY 61-3606 on p38MAPK activity, p38 activation should be analyzed with a phospho-p38 antibody. It would also be informative to determine whether Pa results in Syk activation using a phospho-Syk antibody. Does p38 inhibition reciprocally affect Syk activation?

The reviewer raises and important point. We agree with this in theory, but in the timeframe given for resubmission and despite serious consideration, we concluded that this extra work was not possible, particularly given the additional experimental work required to address this reviewer’s Comment 8, which we judged as enhancing the paper more. We have though amended the manuscript in two ways to address this comment:

- By adding a caveat in line 375-376: “phosphorylation of HSP27, a surrogate of p38MAPK α and β activity”. 

- And by emphasising in the Discussion (line 606-609) that measurement of Pa-induced Syk phosphorylation, and the impact of p38MAPK on this would be of great interest, and should be addressed in future studies. 

[Comment 6]

The protein ladder lanes in Fig 5 are not labeled.

Sorry for this oversight on our part, the protein ladder in figure 5 is now labelled. 

[Comment 7]

The IC50s of RV1088 should be determined and compared with co-treatment with BIRB796/BAY 61-3606.

The IC50 value of 1088 has been calculated and compared with BIRB796/BAY61-3606 when in combination with each other. This has been added at lines 420-423 with the following statement “IC50 values of 2.84 x 10-5 µg/ml (IQR, 2.62-5.05 x 10-5) and 4.14 x 10-5 µg/ml (IQR, 3.04-5.65 x 10-5), respectively (p<0.05, Figs 7A and B). This makes RV1088 six-fold and 600-fold more potent than BIRB796 and BAY 61-3606, respectively, when used in combination against Pa-induced IL-6 from BEAS-2B cells.”

[Comment 8]

RV1088 and/or BIRB796/BAY 61-3606 should also be tested in CFTR (Phe508del) expressing bronchial epithelial cells to evaluate its therapeutic potential.

We thank the reviewer for this comment; we agree that this was an important question to investigate, the data from which could substantially enhance the manuscript. This has therefore been the focus of additional experimental work. We have carried out measurement of Pa-induced IL-6 from a paired bronchial epithelial cell line, CFBE41o-, which express either Phe508del- or WT-CFTR. The results of this are indicated in lines 446 to 459 and confirm that the potencies of BIRB796, BAY 61-3606 and RV1088 are similar in Phe508del and WT CFTR-expressing cells, which are also similar to our original data in BEAS-2B. 

Reviewer #2: 

Minor issues:

[Comment 1]

Please introduce “MAP” abbreviation in abstract.

The MAP abbreviation has now been expanded.

[Comment 2]

Format, Table 1, last row.

The formatting of the last row of table 1 has been fixed to make it a single line.

[Comment 3]

Preferred observe/present data: graphs rather than tables.

Tables 2 and 3 have now been removed and appropriate data added to the text.

[Comment 4}

LPS, Bacterial source and provider/company info?

The LPS was from Pseudomonas aeruginosa 10, this has now been stated along with manufacturer in line 161.

[Comment 5]

Please discuss LPS-Src / MAPK Activation/Potential reasons for negative LPS-induced IL-6 expression in relation to TLRs.

The reason for lack of LPS-induced IL-6 has now been discussed in lines 482 to 483 with the statement “However, they are not well stimulated by certain TLR agonists such as LPS, due to a lack of the co-receptor cluster of differentiation 14 (61).”

Major issues

Pa

[Comment 1]

Strain’s phenotype/ use of Pili, flagella mutants or clinical strains is recommended.

We agree that the use of clinical strains of Pa with varying pheno/ genotypes is extremely important for translational research but this was not within the scope of this study. We consider that a future study would focus on confirming the signalling mechanisms reported in the current manuscript are also observed with clinical strains of Pa. This would require testing of early isolates, acute and chronic strains as well as mucoid and non-mucoid in addition to mutants such as those suggested. Therefore, due to limited time for revision, we decided to conduct this additional work for in a future study, rather than this resubmission. Instead, we stated this in the section of study limitation in Discussion.

[Comment 2]

 Cell exposure consistency: One hour or 2 (line 189)?

The cell exposure time of Pa described in line 189 was optimised for phosphorylation of kinase in the Western blot assays, which is different from the exposure time optimised for cytokine release. As stated in line 201, this two-hour time point was chosen as previous work in our group had shown it to be the optimal time for measuring HSP27 phosphorylation. Gentamicin was not added for Western blots as there was too short time for the bacteria to overgrow and kill the cells. 

[Comment 3]

The role of potential Pa’s Toxins and Proteases should also be addressed for longer incubation times.

Thank you for this comment, we agree that the use of Pa’s toxins and longer time periods would be of great interest. In this study we chose to focus on whole cell infection, but this would be important additional work for our future studies, as discussed in lines 614-618. 

Gentamicin 

[Comment 4]

The purpose for use needs to be mentioned.

A short statement has been added to line 170 stating “gentamicin was added to all wells at a concentration of 100 µg/ml (5) to prevent over-growth of Pa and subsequent epithelial cell death”. 

[Comment 5]

Affects loose or adhered bacteria, which takes several hours, but not cytosolic/intercellular Pa. These issues need to be addressed.

Thank you for this comment, the purpose of the gentamicin addition was to prevent the overgrowth of the Pa and subsequent death of the epithelial cells during the infection period rather than complete elimination of live bacteria. Therefore, we did not carry out further investigation to ensure that all of the bacteria were killed this time. We have addressed this limitation in the discussion in lines 605-609. 

Western blot

[Comment 6]

“kinase phosphorylation assays” may be interpreted as an enzymatic assay than detecting phosphorylated kinase by Western blot.

We agree that the term kinase phosphorylation assays could be misinterpreted. Therefore in line 115 “kinase phosphorylation assays” to “kinase phosphorylation measurement”, in line 196 “Kinase phosphorylation” has been changed to “Western blot of phosphorylated kinases” and in line 371 “A Syk inhibitor blocks Pa-induced HSP27, but not ERK or JNK activity” has been changed to “A Syk inhibitor blocks Pa-induced HSP27, but not ERK or JNK phosphorylation.”

[Comment 7]

Signaling events are switched on quickly and often last a few minutes. As such, western blot analysis of target intermediate molecules and inhibitors effect should have a time/dose curve.

It is true that due the quick and transient phosphorylation of kinases during signalling and potentially different time course with different kinase, a time course study would be ideal. The time point chosen in this study, 2 hours, was based on a time course optimised for measuring HSP27 phosphorylation, which is now referenced in line 202. It has been noted as a limitation of the study in discussion, lines 603-606, in this revised version, that “The use of a single time point in the measurement of kinase-phosphorylation is a limitation, as activation of kinases is transient, and can happen quickly after stimulation. Therefore, future studies would carry out full time course experiments to determine the time of peak phosphorylation of each kinase of interest.” 

MTT assay

[Comment 8]

Need to be performed for Pa, inhibitors, and in combination.

We agree that having a cell viability assay for the compounds alone was a limitation of the study. Therefore additional work has been completed to measure cell viability in the BEAS-2B cells after both compound treatment and Pa infection, with the same time points as for the cytokine release studies. These data are now shown in Figure 2. 

[Comment 9]

Viability protocol is not comparable to assay set up.

We accept that the viability protocol was not exactly the same as the cytokine release assay setup. Therefore, as stated above we have repeated the viability assay to mirror the cytokine studies exactly, and for continuity we have used an MTT assay throughout rather than some MTT and some Resazurin. 

[Comment 10]

Need to observe and report cell morphology (fluorescence microscopy) to confirm MTT data.

We agree that the MTT assay is not a direct measurement of viability, but as a measure of cell metabolic activity is relative to viability. A statement to acknowledge that MTT is a relative measure has been added to line 184-185. We did not have time to complete fluorescent microscopy for this resubmission, but the cells were visually inspected using a light microscope, and there was no sign of changes in cell morphology.

---

## [Decision Letter · Decision Letter 1]

5 Jan 2021

PONE-D-20-23493R1

Pseudomonas aeruginosa induces p38MAP kinase-dependent IL-6 and CXCL8 release from bronchial epithelial cells via a Syk kinase pathway

PLOS ONE

Dear Dr. Coates,

Thank you for submitting your manuscript to PLOS ONE. After careful consideration, we feel that it has merit but does not fully meet PLOS ONE’s publication criteria as it currently stands. Therefore, we invite you to submit a revised version of the manuscript that addresses the points raised during the review process.

Please address the two minor comments made by reviewer #1 and I will accept the paper for publication shortly thereafter.  Thanks, Shawn.

We look forward to receiving your revised manuscript.

Kind regards,

Shawn B Bratton, Ph.D.

Academic Editor

PLOS ONE

Reviewers' comments:

Reviewer's Responses to Questions

**Comments to the Author**

1. If the authors have adequately addressed your comments raised in a previous round of review and you feel that this manuscript is now acceptable for publication, you may indicate that here to bypass the “Comments to the Author” section, enter your conflict of interest statement in the “Confidential to Editor” section, and submit your "Accept" recommendation.

Reviewer #1: All comments have been addressed

Reviewer #2: All comments have been addressed

2. Is the manuscript technically sound, and do the data support the conclusions?

Reviewer #1: Partly

Reviewer #2: Yes

3. Has the statistical analysis been performed appropriately and rigorously? 

Reviewer #1: Yes

Reviewer #2: Yes

4. Have the authors made all data underlying the findings in their manuscript fully available?

Reviewer #1: Yes

Reviewer #2: Yes

5. Is the manuscript presented in an intelligible fashion and written in standard English?

Reviewer #1: Yes

Reviewer #2: Yes

6. Review Comments to the Author

Reviewer #1: Note: I realized the figures had to be individually downloaded to be displayed at full resolution. For some reason when they are imbedded in the PDF the image quality is poor, but otherwise they look fine.

The authors have adequately addressed my concerns with the manuscript, with a few minor exceptions.

1.) The western blot of Fig. 4A has lines between each lane which makes it appear as though it was stitched together from multiple different blots? That would not be suitable for interpretation or publication.

2.) Relatedly, the authors chose to focus on my comment #8, which does add therapeutic relevance to the manuscript. However, in the text they do not address the discrepancies with previously published data. In the discussion the authors cite earlier published data stating that IL-6 release is higher in Phe508del CFTR-expressing cells compared to wild-type, but in their hands it was the opposite.

Reviewer #2: (No Response)

7. PLOS authors have the option to publish the peer review history of their article (what does this mean?). If published, this will include your full peer review and any attached files.

Reviewer #1: No

Reviewer #2: No

---

## [Author Response · Author response to Decision Letter 1]

11 Jan 2021

Dear Dr Bratton,

RE: Revision for PONE-D-20-23493R1 - Pseudomonas aeruginosa induces p38MAP kinase-dependent IL-6 and CXCL8 release from bronchial epithelial cells via a Syk kinase pathway 

We would like to thank the Editor and Reviewer for their comments, which we have carefully considered in preparing this revised manuscript. 

We have addressed the reviewers’ comments point-by-point below. All authors concur with the submission and the material submitted for publication has not been previously reported and is not under consideration for publication elsewhere. Please note, line numbers stated in this letter refer to the file named “Revised Manuscript with Track Changes.”

Reviewer 1

[Comment 1]

The western blot of Fig. 4A has lines between each lane which makes it appear as though it was stitched together from multiple different blots? That would not be suitable for interpretation or publication.

We apologise; the lines were added to make the distinction between lanes clearer, but understand that this could cause confusion. We have, therefore, removed these. 

[Comment 2]

Relatedly, the authors chose to focus on my comment #8, which does add therapeutic relevance to the manuscript. However, in the text they do not address the discrepancies with previously published data. In the discussion the authors cite earlier published data stating that IL-6 release is higher in Phe508del CFTR-expressing cells compared to wild-type, but in their hands it was the opposite.

Thank you for this comment and we accept the discrepancy between the results observed in this study and those from previous studies that are included in the discussion. We have addressed this in lines 488-496 with the statement "The current study did not show greater Pa-induced IL-6 from epithelial cells expressing Phe508del CFTR, compared with WT. This agrees with previous research showing little difference between IL-6 induction from WT and CF primary epithelial cells and cell lines by multiple stimulants (65,66). This however, is counter to the higher Pa-induced IL-6 secretion by Phe508del CFTR expressing cells reported by Berube et al. (22). However, that study used different cell lines, with the CF and WT cells originally from different donors and which were, therefore, not paired. Also, Berube et al. used sterile culture filtrates rather than whole cells, the study, therefore, not being directly comparable to the live infection used in the current study.”

---

## [Editor Report · Decision Letter 2]

13 Jan 2021

Pseudomonas aeruginosa induces p38MAP kinase-dependent IL-6 and CXCL8 release from bronchial epithelial cells via a Syk kinase pathway

PONE-D-20-23493R2

Dear Dr. Coates,

We’re pleased to inform you that your manuscript has been judged scientifically suitable for publication and will be formally accepted for publication once it meets all outstanding technical requirements.

Kind regards,

Shawn B Bratton, Ph.D.

Academic Editor

PLOS ONE
---

## [Editor Report · Acceptance letter]

21 Jan 2021

PONE-D-20-23493R2 

*Pseudomonas aeruginosa* induces p38MAP kinase-dependent IL-6 and CXCL8 release from bronchial epithelial cells via a Syk kinase pathway 

Dear Dr. Coates:

I'm pleased to inform you that your manuscript has been deemed suitable for publication in PLOS ONE. Congratulations! Your manuscript is now with our production department. 

Kind regards, 

on behalf of

Dr. Shawn B Bratton 

Academic Editor

PLOS ONE